Taxonomic and chemical assessment of exceptionally abundant rock mine biofilm

Tomczyk-Żak Karolina 1
Szczesny Paweł 2 3
Gromadka Robert 4
Zielenkiewicz Urszula ulazet@ibb.waw.pl 1
1 Department of Microbial Biochemistry, Institute of Biochemistry and Biophysics, PAS , Warsaw , Poland
2 Faculty of Biology, Institute of Experimental Plant Biology and Biotechnology, University of Warsaw , Warsaw , Poland
3 Department of Bioinformatics, Institute of Biochemistry and Biophysics Polish Academy of Sciences , Warsaw , Poland
4 Laboratory of DNA Sequencing and Oligonucleotides Synthesis, Institute of Biochemistry and Biophysics, PAS , Warsaw , Poland
Souza Valeria
Electronic publication date: 2017 Aug 15
Publication date: 2017
Volume: 5
Electronic Location ID: e3635
Received 2017 Apr 13; Accepted 2017 Jul 11
Copyright: ©2017 Tomczyk-Żak et al.
Copyright year: 2017
Copyright holder: Tomczyk-Żak et al.
License: This is an open access article distributed under the terms of the Creative Commons Attribution License, which permits unrestricted use, distribution, reproduction and adaptation in any medium and for any purpose provided that it is properly attributed. For attribution, the original author(s), title, publication source (PeerJ) and either DOI or URL of the article must be cited.
License URL: https://creativecommons.org/licenses/by/4.0/

Keywords: Biodiversity, Metagenomics, Mine, Biofilm

Funding: The authors received no funding for this work.

==============================
Background

An exceptionally thick biofilm covers walls of ancient gold and arsenic Złoty Stok mine (Poland) in the apparent absence of organic sources of energy.

Methods and Results

We have characterized this microbial community using culture-dependent and independent methods. We sequenced amplicons of the 16S rRNA gene obtained using generic primers and additional primers targeted at Archaea and Actinobacteria separately. Also, we have cultured numerous isolates from the biofilm on different media under aerobic and anaerobic conditions. We discovered very high biodiversity, and no single taxonomic group was dominant. The majority of almost 4,000 OTUs were classified above genus level indicating presence of novel species. Elemental analysis, performed using SEM-EDS and X-ray, of biofilm samples showed that carbon, sulphur and oxygen were not evenly distributed in the biofilm and that their presence is highly correlated. However, the distribution of arsenic and iron was more flat, and numerous intrusions of elemental silver and platinum were noted, indicating that microorganisms play a key role in releasing these elements from the rock.

Conclusions

Altogether, the picture obtained throughout this study shows a very rich, complex and interdependent system of rock biofilm. The chemical heterogeneity of biofilm is a likely explanation as to why this oligotrophic environment is capable of supporting such high microbial diversity.

Introduction

The mutual interactions of microbes with surroundings result in both environmental and microbial community changes. The significant role of active microorganisms in element biotransformations and biogeochemical cycling, metal and mineral transformations has been recognized (Gadd, 2010). The structure of microbial communities in a particular environment depends on specific physical and chemical conditions: humidity, pH, temperature, salinity, concentration of oxygen, heavy metals and other toxic compounds, and the availability of electron acceptors and carbon sources.

Sub-aerial biofilms, ubiquitous on the solid surfaces exposed to the atmosphere, form relatively stable miniature ecosystems that contribute to the weathering of natural rocks and human constructions (Gorbushina, 2007). Similarly, the diversity and abundance of epi- and endolithic prokaryotic communities in the deep sea also positively correlates with the extent of rock alteration (Santelli et al., 2009). However, only sporadic development of microorganisms on the surfaces in the form of abundant biofilms has been documented. In most cases these were the micro-colonies on prehistorical paintings in caves (Schabereiter-Gurtner et al., 2002a; Portillo, Gonzalez & Saiz-Jimenez, 2008; Portillo, Gonzalez & Saiz-Jimenez, 2009; Schabereiter-Gurtner et al., 2002b; Schabereiter-Gurtner et al., 2004) or populations colonizing historical monuments (Gorbushina et al., 2002; Zimmermann, Gonzalez & Saiz-Jimenez, 2006). In turn, in caves with acidic pH, a population of biofilms called “snottites” is limited only to a few microbial species (Macalady, Jones & Lyon, 2007).

Contrary to caves, mines are not considered a “natural” environment. An interest in the biodiversity of mines is associated with the strong negative effects of mining activities on the environment, particularly on ground and surface waters. Most well-studied mine ecosystems inhabit both rock surfaces as well as a mining waste such as slag heaps and acidic, metal-rich waters referred to as “acid mine drainage” (AMD) or “acid rock drainage” (ARD). AMD environments are the result of the accelerated oxidation of exposed minerals after exploitation of metal ores and coal. Usually such environments are rich in sulphur compounds and are generally characterized by high concentrations of metal ions (e.g., aluminium, copper, zinc, manganese, arsenic, and especially iron) and high temperatures, which are the result of strongly exothermic oxidation reactions of these compounds. The amount of organic matter is generally low (<20 mg/l). In the majority of acidic environments the diversity of the microbial population is low, comprising one or a few dominant species (Johnson & Hallberg, 2003; Baker & Banfield, 2003), which are usually chemolithoautotrophic microorganisms that carry out oxidation and reduction reactions of ferrous and sulphur ions, acidophilic heterotrophs and/or facultative heterotrophs. The heterotrophs are indirectly involved in the dissolution of minerals by using organic compounds (e.g., organic acids) produced by autotrophic organisms and detoxify the autotrophic bacteria environment. They represent both the Bacteria and Archaea domains.

Few studies have examined cave or mine environments with the neutral or alkaline pH (Pasić et al., 2010; Labrenz & Banfield, 2004; Lin et al., 2006), as opposed to the acidic environment of mines resulting from chemical procedures applied during the mining process. Species composition in neutral or alkaline environments differs from the acidic ones and is generally more diverse. Moderately alkaline pH (7.4.–8.1) is a trait of the Złoty Stok gold and arsenic mine geochemistry. The Złoty Stok mine in Poland has been exploited for gold since the 12th century and for arsenic since the 18th century. Mining ceased approximately 60 years ago, but its arsenic deposits are still among the largest in the country and are the primary source of arsenic contamination of surface and ground water in the area.

The microbiology of this particular environment has been investigated in the context of arsenic cycling. It was shown that both types of microbial communities inhabiting the mine, mats and rock biofilm, can contribute to the dissemination of arsenic into mine water. Metagenomic studies revealed high arrA and aioA genetic diversity in these communities (Drewniak et al., 2012; Tomczyk-Żak et al., 2013). Bacteria isolated from this mine can directly and indirectly contribute to the mobilization of arsenic from minerals into the water and sediments (Drewniak et al., 2010; Drewniak et al., 2014). However, the details of the transformation of immobilized arsenic into its soluble form are still to be elucidated.

In previous studies, the biodiversity of rock biofilm at this particular spot was analysed by 16S rRNA gene clonal analysis (Tomczyk-Żak et al., 2013). Here we present the results of the biodiversity assessment of the Złoty Stok mine biofilm obtained in culture-dependent and -independent manners (using pyrosequencing method) and discuss the possible interactions of biofilm microorganisms with the environment. The biofilm, despite being present in nutritionally poor niches, has unexpectedly high diversity and complexity. This is likely explained by spatial heterogeneity shown by SEM-EDS and X-ray analyses.

Materials and Methods

Site description and sample collection

The former arsenic and gold mine Złoty Stok (50°26′N, 16°52′E) is located in the Sudety Mountains in southwest Poland. The mine lies within metamorphic rocks of the geological tectonic zone of the western Sudetes composed of mica and mica-quartz schists, amphibolites, leptynites, gneisses, serpentinites and crystalline limestone with characteristic polymetallic mineralisations (Przylibski, 2001).

On the rock surface in the deepest section of the Gertruda Adit, a natural microbial biofilm is formed (Fig. 1). In this part of the mine, the air is characterized by a stable temperature of 10.4–11.1 °C, a reduced concentration of oxygen (17.2%), and levels of arsenic hydride in the range of 1.52–3.23 mg/m3 (Drewniak & Styczek, 2008). Apart from CO2 and N2, other gaseous components in the Adit, including simple organic compounds (aliphatic and aromatic hydrocarbons, volatile alcohols, aldehydes and acids), occur in trace amounts. The rock in the Gertruda Adit contains a variety of arsenic-bearing minerals, including loellingite (FeAs2) and arsenopyrite (FeAsS) and other minerals containing primarily iron, lead, zinc and copper, generally in the form of different sulphides.

Figure 1 Rock biofilm from Złoty Stok gold and arsenic mine.

(A) Photograph of biofilm in situ; (B) Micrograph of unstained biofilm in phase contrast; (C) Micrograph of cotton-blue stained biofilm presenting different bacteria, including Actinobacteria; (D, E). Scanning electron micrographs of an unprocessed rock biofilm sample (D: ESEM magnification 50×; E: FESEM, magnification 5,000×). Arrows indicate mineral particles.

In November 2007, samples of biofilm were carefully collected from the walls in the end section of the Gertruda Adit, 2 km from the entrance. Portions of biofilm of approximately 40 g were aseptically cut from the rock directly into sterile 50-ml tubes. These samples were stored at 4 °C and processed within 24 h of collection. For cultivable bacteria isolation, additional biofilm sampling was performed in June 2008.

DNA extraction

Total biofilm DNA extraction was performed as described by (Tomczyk-Żak et al., 2013). Briefly: biofilm samples were homogenized using glass beads in the presence of the extraction buffer by shaking in a MiniBead-Beater (Bio-Spec Products, Bartlesville, OK, USA), subsequently incubated at 37 °C for 30 min with a mixture of lysozyme (10 mg/ml) and zymolyase (0.05 mg/ml), lysed by addition of 20% SDS and frost-thaw repetitive cycles, after which DNA was extracted with chloroform/isoamyl alcohol (24:1) and isopropanol precipitation. The DNA was further purified using the NucleoSpinTissue kit (Macherey-Nagel, Düren, Germany). Three independent biofilm samples (2 g each) were processed and finally pooled for all PCR amplifications.

Bacterial genomic DNA was obtained after enzyme treatment (the mixture of lysozyme, lysostaphin and mutanolysin) followed by the use of a commercial isolation kit (Genomic Mini A&A Biotechnology, Gdynia, Poland).

Amplicons preparation and 454-pyrosequencing

Separate procedures were carried out to determine the bacterial, actinobacterial and archaeal community composition. Fragments of the correspondent 16S rRNA genes were amplified from biofilm total DNA using Phusion polymerase (Finnzymes) and appropriate primers that were fused to Roche-suitable MID oligonucleotides: for Bacteria16S rDNA—universal MB-16SrF, M6-16SrR; for Actinobacteria M10- 337F, MB-1159R; for Archaea—M8-A21F, MB-1204R; and in nested PCR—M8-A21F, MB-518R (Table S1). Amplifications were performed using optimized thermal cycles (Table S2). PCR products were purified with a NucleoExtract II kit (Macherey-Nagel). The concentration and quality of the PCR products were assessed with Picogreen staining and a ChipDNA Bioanalyzer, and equal amounts were sequenced using a Roche GS FLX Titanium sequencer with a standard 454 protocol.

Sequence data from the GS FLX Titanium run has been deposited at the NCBI Short Read Archive (SRA) under project ID of SRP093827.

Isolation and identification of bacteria

Bacteria were isolated from the biofilm samples (several dozen, 6 g in total), which were suspended in 0.8% NaCl and homogenized with glass beads (Merck 1kb) by shaking for 60 min at room temperature. After centrifugation (10 min at 8,000 rpm) the pellets were re-suspended in a small volume of 0.8% NaCl and spread in parallel over several types of solid media: LB (Luria Bertani, Difco), selective AIA (Actinomycete Isolation Lab-Agar, Biocorp), SC (starch casein; g/L: starch 10, casein 0.3, KNO3 2, NaCl 2, K2HPO42, MgSO4×7H2O 0.05, CaCO3 0.02, FeSO4×7H2O 0.01, pH 7–7.2), and GYA (glycerol agar; g/L: glycerol 5, yeast extract 2, K2HPO4 0.1, peptone 25), each containing 5% glycerol and supplemented with 50 µg/ml of cycloheximide. The plates were incubated at 14 °C or 20 °C, in darkness, for up to two weeks. Part of the plates were prepared and incubated under anaerobic conditions (Whitley A35 workstation). Morphologically different isolates were transferred onto respective selective- and LB-agar plates to obtain pure cultures.

The individual strains were selected using the multi-temperature single-strand conformation polymorphism (MSSCP) method of genetic profiling by choosing unique MSSCP profiles of their PCR-amplified (Tables S1 and S2) V3 fragments of 16S rDNA, as described in Tomczyk-Żak & Kaczanowski (2012). Briefly, in the applied colony, PCR-technique colonies of cultured biofilm strains were suspended in 100 µl of lysis buffer, boiled for 5 min and centrifuged. Equal volumes of sterile cold water were added to the supernatants, and these were used as templates in PCR for amplification of 160-bp long fragments of 16S rRNA gene sequences.

In the case of difficult-to-lyse strains, the isolated genomic DNA of the cells (see above) was used.

PCR amplification and sequencing of 16S rRNA genes of cultured biofilm bacteria

To sequence the 16S rRNA genes of cultured bacteria selected by MSSCP, fragments of 16S rDNA were amplified using Paq Polymerase (Stratagene) and the universal primers 27F and 926R (Tables S1). Genomic bacterial DNAs (app. 3.7 ng/µl) were the templates in the optimized thermal cycle reactions (Table S2). PCR products were purified with NucleoExtract II kit (Macherey-Nagel) and sequenced using the same primers on an ABI3730/xl Genetic Analyzer (Applied Biosystem) at the Laboratory of DNA Sequencing and Oligonucleotide Synthesis, IBB, PAS. Sequences were assembled using Phred, Phrap and Consed Linux programs (Ewing et al., 1998; Ewing & Green, 1998). The final sequences were compared with available nucleotide databases GenBank (NCBI) using BLAST (Altschul et al., 1997) (http://blast.ncbi.nlm.nih.gov/Blast.cgi). The sequences were deposited in GenBank with the accession numbers GU213114 –GU213156.

Analysis of biodiversity

The reads from pyrosequencing were processed using MOTHUR (Schloss et al., 2009) software according to Schloss Standard Operating Procedure described on MOTHUR’s wiki (accessed 07.2014). Sequences were de-noised, filtered based on their quality and aligned using a Silva-compatible database of SSU rRNA genes provided by MOTHUR’s authors. Potential chimeras were identified using UChimie (Edgar et al., 2011) algorithm and subsequently removed. Taxonomic assignments were completed with the RDP classifier (Wang et al., 2007). Operational Taxonomic Units (OTUs) were defined at the threshold of 98% of identity. Data from Tomczyk-Żak et al. (2013) (added under label: Clones) were analysed jointly using the same protocol. The mean length of sequences before processing in each group was: 1,094 (clones), 131 (archaeal primers), 219 (generic primers) and 113 (actinobacterial primers), which was the basis for picking the higher threshold of OTU definition.

Functional analysis

Functional capabilities of the biofilm have been assessed with the METAGENassist server (Arndt et al., 2012), which maps phenotypes of known microbial species onto genera identified during the taxonomic classification of reads. Final OTU sequences were used in this step.

Scanning Electron Microscopy (SEM)

Unprocessed samples of the biofilm were examined directly by environmental (ESEM—FEI QUANTA 200) and field-emission (FESEM—JSM 7401F) scanning electron microscopy.

The elements C, O, S, As and Fe were detected in unprocessed samples (dried only) of the biofilm by SEM (JEOL JSM-6380LA) coupled with an energy dispersive X-ray spectroscope (EDS). Analysis was performed for 17 h at an accelerating voltage of 20 kV under low vacuum (40 Pa).

The mineral composition of the bedrock was determined on a polished surface by SEM-EDS (JEOL JSM-6380LA) at a 10-mm working distance with 100 s of live time. The chemical composition of the interface was determined in thin layer embedded in resin by both, X-ray SEM-EDS (JEOL JSM-6380LA) and photoelectron spectroscopy (XPS) using Cameca SX100 operating with electron beam of 15 keV at 10–40 nA. X-ray transition energy measurements were realized using WDS with PAP correction. Analyses were done in the Joint Laboratory of Microanalysis of Minerals and Synthetic Substances, Faculty of Geology, Warsaw University.

Results

Structure of biofilm

Exceptionally thick (up to 3 cm) gelatinous biofilm covers several dozen square meters of the rock wall in the Gertruda Adit. It was described in detail by Tomczyk-Żak et al. (2013), and here we briefly recap its main features.

A single, separate grop of biofilm occupies 5–10 m2. The overall morphological appearance of the biofilm is variable in terms of colour, smoothness, consistency, moisture content, thickness and tubercle dimensions, yet the internal structure revealed by SEM methodology shown in every case the lattice structure of the matrix with vast empty or lower electron density spaces (Figs. 1D and 1E). This heterogeneous, abundant, hydrated, inorganic matrix encloses both bacteria and small mineral particles. Most of bacteria seem to occupy well-defined areas. Different microscopy techniques uncovered a variety of bacterial shapes. Visualized bacteria represent compact uniform communities as well as morphologically diverse clusters (Fig. 1C). Further analyses were performed on three distinct layers of the sample—the biofilm (layer 0), an interface between biofilm and underlying rock (layer 1), and finally the solid bed under the biofilm (layer 2: see Fig. 2 schematic).

Figure 2 Diagrammatic cross-section of the rock biofilm test sample.

Pyrosequencing

DNA extracted from the biofilm was processed as described in the Methods. Then, segments of ribosomal 16S rRNA genes were amplified in three independent reactions using sets of primers targeted at Archaea, Actinobacteria and an universal set (Table S1). With a similar amount of sequenced DNA, we obtained an unequal number of reads from each sample (see Table 1). The archaeal sample seemed to be sequenced most deeply as it had a high number of reads and the lowest number of OTUs. This is reflected in the highest value of Good’s coverage estimator and the rarefaction curve approaching horizontal line (Fig. S1, curve B,). The other two samples appear to be under-sequenced, as the Good’s coverage value for them is only around 0.2. In the case of the actinobacterial sample, this is mainly the issue of the low number of good quality reads, since the value (100.6) of inverse Simpson parameter (also called Simpson’s Reciprocal Index, 1/D) suggests that we have captured the majority of biodiversity in that sample. As for the universal primers the issue appears to be related to the extremely high diversity revealed. The inverse Simpson parameters are two orders of magnitude larger than in the other two cases, therefore it is very likely that further sequencing would provide additional biodiversity.

Table 1 Basic statistics of three sequenced samples obtained using the MOTHUR software.

OTUs were defined at the threshold of 0.02 dissimilarity.

Primer set	No: sequences/ OTU	Good’s coverage	Chao1	Inverse Simpson	ACE	Shannon	
Archaeal unique sequences	14,157	0.88	8,088	164.8	13,783	6.2	
Actinobacterial unique sequences	327	0.21	3,533	100.6	25,955	5.3	
Universal unique sequences	29,427	0.20	33,3172	22,150	11,09221	10.0	
Archaeal OTUs	2,574	0.91	5,289	2.3	8,917	2.8	
Actinobacterial OTUs	269	0.48	818	21	2,403	4.6	
Universal OTUs	20,336	0.38	15,5644	3,834	47,1427	9.4	

Figure 3 Summary of taxonomic classification obtained from pyrosequencing of three samples based on classification of 16S rRNA reads with the RDP classifier.

Clades with more than 10 reads found in each primer set were marked with outer bars in blue, pink and green for universal, archaeal and actinobacterial primers, respectively. The number of sequences in each data set is the same as the number of unique sequences from Table 1. For a comparison, clones from Tomczyk-Żak et al. (2013) were added as the red bars. Branches of more than 1% of total reads were marked with black dot.

Taxonomic assignments

There is some overlap between taxonomic coverage of the samples (Fig. 3), but overall primers targeted at specific taxonomic groups had a relatively narrow and correct focus. Qualitative overlap in the OUT assignment is shown in Fig. S4. Archaeal OTUs could be identified only by the primers targeted specifically at this taxonomic group. The majority of assignments of actinobacterial primers were from that phylum. This is in agreement with studies analysing the impact of PCR amplification on the proportion of rare biosphere in sequencing of microbial communities (Gonzalez et al., 2012).

An overview of taxonomic assignments across all samples is shown in Fig. 3. The tree represents all the genera identified by their taxonomic assignments, but for clarity of mapping (the outer layer of the bars) has been shown only for the genera that had at least 10 reads per primer set. Detailed breakdowns of taxonomic assignments of reads and OTUs are available in (Tables S3 and S4, respectively).

The analysed sequences were classified into twenty four phyla (partially depicted in Fig. 3): Crenarchaeota (1.3%), Euryarcheota (6.2%), Acidobacteria (1.0%), Actinobacteria (1.8%), Aquificae (0.1%), Bacteroidetes (4.3%), Chlamydiae (0.2%), Chlorobi (<0.1%), Chloroflexi (2.3%), Deferribacteres (<0.1%), Deinococcus-Thermus (<0.1%), Firmicutes (1.6%), Fusobacteria (<0.1%), Gemmatimonadetes (1.3%), Lentisphaerae (<0.1%), Nitrospira (0.1%), Planctomycetes (0.5%), Proteobacteria (78.3%), Spirochaetes (<0.1%), Synergistetes (<0.1%), Tenericutes (<0.1%), Thermodesulfobacteria (0.2%), Thermotogae (<0.1%) and Verrucomicrobia (0.3%). Many archeal sequences were assigned to several known methanogenic phyla, such as Methanomicrobia, Methanococci or Methanopyri (4.3%, 0.9% and 7.2% of all sequences assigned to Archaea, respectively). Methanomicrobia were also found in the bottom mat from this mine by the Drewniak group of the Laboratory of Environmental Pollution at the University of Warsaw (unpublished data, available at MG-RAST ID mgm4554870.3). The most highly represented phyla in the Bacteria domain were Proteobacteria, Bacteroidetes, Actinobacteria, Chloroflexi, Firmicutes and Acidobacteria (Table S3). The structure of the biofilm community was dominated by bacteria belonging to α-Proteobacteria (62.3% of all reads), especially to Rhizobiales (almost 8% of all reads—43,911—were assigned to that clade). Microorganisms representing the Hyphomicrobiaceae, Beijerinckiaceae, Rhodospirillales and Methylobacteriaceae were the most abundant groups of α-Proteobacteria in the biofilm population. The rest of the Proteobacteria sequences were assigned to δ-Proteobacteria (10.2%), γ-Proteobacteria (5.2%), and β-Proteobacteria (0.6%) (listed in order of abundance). The Actinobacteria phylum was represented mainly by sequences from the Actinomycetales and Solirubrobacterales orders, while Bacteroidetes was represented by the Sphingobacteriales order and Flavobacteria class.

Assuming that the PCR with different primers introduced a heavy bias in the assessed structure of the communities of all three samples (Xu et al., 2011), we did not attempt to statistically assess relative abundances between Archaea, Actinobacteria and other taxonomic groups.

Phenotypic analysis of taxonomic groups

On the basis of taxonomic assignment, we performed an analysis of the phenotypic potential using METAGENassist tool (see Methods). The results are summarized in (Fig. S2). This analysis suggests that almost all identified genera for which phenotypic information was available represent aerobic species, while the majority of them are autotrophic. There is a substantial number (over 60% in each group, according to METAGENassist) of free-living species. The main group of biofilm microorganisms were connected with the nitrogen cycle: ammonia oxidizers able to carry out the first step of nitrification, and nitrite reducers, engaged in partial de-nitrification. Among them Planctomycetes are, most probably, responsible for transformation of mineral nitrogen in this environment. Diazotrophs assimilating atmospheric nitrogen were also frequently found. Other identified microorganisms were sulphur metabolizing: sulphate reducers, sulphur oxidizers, and metabolized other S-compounds. This finding was to be expected due to the large presence of sulphur minerals such as pyrite (FeS2), chalcopyrite (CuFeS2), arsenopyrite (FeAsS) and dimethyl disulphide (C2H6S2). What was surprising was the lack of identified by METAGENassist microorganisms capable of the reduction/oxidation of iron. Their presence was expected due to the high concentration of iron-containing minerals, the identification of such microorganisms in clonal analysis and measured elevated activity of the biofilm siderophores (Tomczyk-Żak et al., 2013). A detailed analysis of the list of identified taxa indicated that a vast number of organisms can potentially be involved in iron redox transformations (e.g., bacterial genera: Acidiphilium, Ferrithrix, Acidimicrobium, Anaeromyxobacter, Geothrix, Shewanella and archeal genera: Ferroglobus, Geoglobus). It is notable, that among the cultivated strains, Paenibacillus and Stenotrophomonas represent genera able to reduce and oxidize iron, respectively.

It is also noteworthy that, according to METAGENassist analysis, the biofilm microbiome taxonomic profile suggests a wide range of metabolic capabilities, including, among others, dehalogenation, degradation of aromatic hydrocarbons or chitin as well as metabolism of pollutants and other toxic compounds. It is suspected that two groups of bacteria abundant in the analysed sample, rhizobia and actinobacteria, already known for the presence of species capable of dealing with complex compounds, are responsible for that result.

Some taxonomic branches might contain bacterial species capable of arsenic mobilization (e.g., the genera Bosea, Microbacterium, Clostridium and Leptospirillum).

Cultured bacteria

A total of 52 different bacterial strains from Złoty Stok biofilm were isolated using different media, and they were subsequently cultured. Their partial 16S rRNA genes were amplified and sequenced. Taxonomic assignment was completed using the same approach as with reads from pyrosequencing, i.e., MOTHUR implementation of RDP classifier. Results presented in Fig. 4 show 38 genera belonging to four phyla: Actinobacteria, Proteobacteria, Firmicutes and CFB (the last two groups are represented by few genera). Bacteria that could not be assigned to the genus level (but classified to the family level) may represent species not yet identified. Interestingly, strains of all four recognized taxonomic groups were isolated on all used media with the exception of the CFB group, representatives of which were not isolated on LB.

Figure 4 Taxonomic classification of cultivated strains from rock biofilm based on classification of 16S rRNA genes with the RDP classifier.

The topology reflects taxonomic classification of RDP database. Clades of phyla are highlighted in gray. Different cultivation media are marked with gray rectangles.

Among 38 cultivable genera, 25 were also identified in a metagenomic approach. However, a comparison of phenotypic potential between different groups (as assessed by METAGENassist, see Fig. S2) indicates that the growth conditions used for bacteria isolation (see Methods) were not overly selective. Ratios of genera with specific phenotypic traits (such as the ability to use different energy sources, oxygen requirements, metabolic capability or biotic relationship) are similar between the cultured bacteria and the samples derived using Actinobacteria-specific and generic primers.

Elemental analysis

The analysis of solid bedrock under the biofilm was performed using SEM-EDS. Previous geological studies of the Złoty Stok territory (Wierzchołowski, 1976; Przylibski, 2001; Muszer, 2011) indicated that we should expect minerals such as plagioclase, quartz, micas (biotite, muscovite), apatite, monazite, K-feldsparts and ore minerals (pyrite, pyrrothite, chalcopyrite, sphalerite, loellingite, arsenopyrite). In the particular area analysed, we have found majority of these minerals (Fig. S3), except for most of the ore minerals (only pyrite was found) and arsenic minerals.

The interface between rock and biofilm was analysed using both, SEM-EDS and X-ray techniques. The summary of quantitative analysis is present in Fig. S3C. In essence, the qualitative picture revealed by this analysis did not differ much compared to solid bed. The presence of many secondary minerals (e.g., zircon, rutile, monazite, xenotime) and sulphide ore minerals (pyrrothite, chalcopyrite, sphalerite) was detected. However, we repeatedly observed interesting structures of the intrusions of pure noble metals (silver and platinum) in elemental form (Fig. 5). They did not look like typical crystal intrusions, raising a question of their biogenic origin. Arsenic intrusions were not observed.

Figure 5 SEM-EDS analysis of the interface between rock and biofilm.

Individual peak numbers from (C–F) correspond to spot numbers visible in the (A, B). Two columns represent two main metals: (A) platinum, (B) silver. Irregular diffused (non-crystal) metal intrusions are seen clearly on (A).

Elemental analysis of the biofilm layer was performed using the SEM-EDS mapping technique (see Methods). Distribution of several elements, such as carbon, sulphur, oxygen (together with silicon), arsenic and iron, were analysed on the biofilm surface (Fig. 6). Some elements, such as carbon, sulphur and oxygen/silicon were distributed unevenly, and their location was highly correlated. Arsenic and iron were distributed more evenly, with a tendency to concentrate in mineral particles. Notably, arsenic was nearly absent in areas of high carbon concentration, while the concentration of iron increased in the areas where carbon and sulphur were present. As stated above, the biofilm contains numerous mineral intrusions of different sizes in its volume (shown in Fig. 1).

Figure 6 Mapping of elements in the rock biofilm sample.

SEM-EDS elemental maps showing C, S, O, As and Fe distribution in the biofilm sample. In the oxygen/silicon panel oxygen is presented by gray colour and silicon by black colour. Arrows indicate large mineral intrusions.

Discussion

In this study we analysed the exceptionally abundant biofilm developing in extremely oligotrophic environment on the walls of the Gertruda Adit of the Złoty Stok mine. Both elemental analysis and biodiversity assessment revealed unexpected diversity and complexity. Presented work sheds new light on several areas.

Rich ecosystem of oligotrophs

The microbiota of this ancient gold mine probably represents indigenous microorganisms from the ore and fracture water, plus others that were brought in with spruce wood beams and other timbers, and also by the mine workers of fifty years ago and earlier. However, the present environmental conditions (especially the high arsenic concentration) are more favourable for indigenous microorganisms. These conditions have led to a level of bacterial diversity that is comparable with that of partially reclaimed tailings or old, inhabited caves (Shannon index 8–12) rather than drilled rock cores or newly formed or opened caves (Shannon index 1–5).

We have assessed biodiversity using pyrosequencing of fragments of the 16S rRNA gene, which were amplified using primers targeted at Archaea and Actinobacteria, and using universal primers. As a result, a large number of additional taxonomic groups has been revealed, substantially enriching (50 taxonomic families more, that is almost 30%) the assessment obtained using universal primers only. This alone is not surprising, as the bias coming from the choice of primers has been a topic of intensive studies over the last few years (Bergmann et al., 2011; Cai et al., 2013). The high biodiversity (Shannon index reaching 10) was reflected in experiments with cultured bacteria: major taxonomic groups revealed by pyrosequencing were later identified within cultured bacteria. In total we have found almost 4,000 OTUs. The majority of organisms are from Proteobacteria, Actinobacteria and the archaeal genera of methanogenic capabilities. However, there is very likely a long tail of strains capable of perusing complex biofilm structure for growth, which could be revealed by deeper sequencing.

Independently, we cultured 52 microbial species and sequenced their near full-length 16S rRNA genes. As expected, there was an overlap in the biodiversity assessment between the culture-dependent and culture-independent analyses. Speculating based on the taxonomic assignments, it seems that more than half of the revealed community organisms possess small genomes, 1–2 Mb in size, suggesting their specialized rather than versatile metabolism (Giovannoni et al., 2005). Hence, the mutual interactions between biofilm organisms seem to be very complex. We could not reliably identify known examples of syntrophic interactions between pairs of bacterial species. However, functional analysis (which infers from cultivable, well-studied species) showed many potential energy sources, further supporting the hypothesis that many complementary functional interactions occur between species inhabiting the biofilm. Based on metagenomic analysis, we can distinguish three types of functional interactions (involving nitrogen, iron together with sulphur, and finally methane) between biofilm microorganisms and the environment as shown schematically in Fig. 7. Some reactions were confirmed by analysis in the previous work (Tomczyk-Żak et al., 2013). The major sources of energy appear to be some nitrogen compounds, which is reflected in increased levels of nitrogen and nitrous oxide. These are likely supplemented by sulphur compounds, given that sulphur is an abundant element in minerals present in the habitat. Another interesting capability is methano/methylotrophism, as it complements the methanogenic activity of bacteria present in bottom sediments of this mine (Drewniak et al., 2012). Obviously, the majority of microorganisms will be capable of perusing iron ions, as the energy sources, given the high concentration of iron-containing minerals, but this process is missing in the results of METAGENassist analysis, most likely due to limitations of available phenotypic databases (e.g., BacMap https://www.ncbi.nlm.nih.gov/pubmed/22135301, the data source for METAGENassist, contains less than 2,000 genomes). Numerous members of genera showing iron-capturing capabilities have been detected in clonal analysis (Tomczyk-Żak et al., 2013). As mentioned above, among cultivated strains, two were able to utilize Fe ions.

Figure 7 Model showing putative participation of rock biofilm microorganisms in the transformation of chemical compounds in the Złoty Stok mine.

METAGENassist predictions from this study overlap with data from our previous report. Undetected transformations are implied by literature data. Chemical transformation: (A) nitrogen; (B) iron and sulphur; (C) methane.

The presence of bacterial species in the Złoty Stok mine that are highly resistant to arsenic has been known for a few years (Drewniak & Styczek, 2008). The key genes involved in such a process, i.e., arsenite oxidase or arsenate reductase, were detected in other studies by PCR amplification (Tomczyk-Żak et al., 2013). However, we were unable to culture a bacterial species that uses arsenic as an energy source. Also, so far, there has been no clear answer as to which species from the rock biofilm is capable of arsenic mobilization, although Drewniak and co-workers identified and cultured such a species in the bottom mat of the mine (Drewniak et al., 2010; Drewniak et al., 2012; Drewniak et al., 2014).

Given that arsenic has an even distribution in the biofilm, it is plausible that most of the biofilm inhabitants do not use arsenic compounds as energy sources. Some studies, based on PCR amplification of arsenite oxidase or arsenate reductase genes, suggest that these enzymatic features are much more common than previously thought (Oremland & Stolz, 2003). However, given the toxicity of arsenic, it is more plausible that most of organisms would prefer to use other sources of energy over inorganic arsenic.

Mobilization of other elements

Precious metals such as gold, silver and platinum typically exist in the rocks as part of iron ore. In the Złoty Stok mountain area, polymetallic mineralisations have been noted (Przylibski, 2001). Metal inclusions, if present, have the characteristic shapes of grains of several types. Precious metals are present in the interface between the solid bed and the biofilm; however, the shape of intrusions is strikingly unusual for all elements analysed. It is tempting to suggest that these grains are of biogenic origin. Bacteria, perusing iron compounds from an ore, also capture the precious metals. As these elements are not utilized in any way, they might be deposited in an amorphous form. Also, the co-occurrence of methanotrophs and methanogenes can substantially influence the metal and metalloids mobilization. Choi et al. (2006) showed that methanotrophy may play a role in either the solubilization or immobilization of many metals in situ. Several organisms, e.g., Streptomyces, Desulfovibrio and Variovorax from among the genera described as able to recover precious metals, have been detected in the studied biofilm. Bacterially induced mineralization processes become well recognized and led to the use of microorganisms for recovery of precious metals, especially gold, silver and platinum (Das, 2010).

Conclusions

Limited resources in the environment should limit biodiversity according to Gause’s law. Competing for scarce sources of energy leads typically to extinction of the weaker species. However, certain habitats support high biodiversity despite limited resources, and this situation is called “the paradox of the plankton” (Hutchinson, 1961), as plankton was the first example of this kind of environment. The rock biofilm present in the Złoty Stok mine is another example of such a paradox. While the environment is extremely oligotrophic and has no light sources, the observed community exhibits extremely high richness and complexity. This cannot be explained by large spatiotemporal dynamics of the population, as the bacteria in the biofilm are mostly immobilized. However, given the spatial difference in concentration of arsenic and other metals across the analysed layers, it is plausible that there is a gradient of inorganic compounds including trace elements within the biofilm structure, which support the heterogeneity of the community. Additional support might come from the water; organic carbon from the water residing on the biofilm is below 160 mg/L. Chemical analysis indicated that the water samples from the Gertruda Adit are likely to be of atmospheric origin. While there is no visible dripping water, the moisture of the walls most likely has an external origin. Therefore, the outermost layers of the biofilm have access to additional organic compounds at low concentrations.

In conclusion, we extended our previous investigations of microbial biofilm composition using both culture dependent and culture independent methods. Furthermore, we correlated microbial community structure to mineral and elemental composition of both, biofilm and rock, and based on the analysis of taxonomic assignments, we predicted biogeochemical transformation pathways. The prediction agrees with literature data, but in the absence of direct experimental evidence remains a rough speculation.

The special features of this community are its abundance and very high diversity. As expected, amplicons pyrosequencing with specific primers uncovered the presence of archaeal species and abundant actinobacterial population which had escaped from previous clonal analysis.

In this report, the ecological “paradox of plankton” is postulated as the explanation of both the extremely high microbial diversity and the abundance of the mine rock biofilm developing in an oligotrophic environment. The cycling energy flow of most prominent minerals in the absence of evident influx of organic matter could explain the relationship between geochemical processes and the activity of microorganisms in this peculiar anthropogenic environment.

Supplemental Information

Figure S1 Rarefaction curves for 16S rRNA sequences obtained with different primer set

(A) Actinobacteria, (B) Archaea, (C) universal.

Curves were calculated at 2% evolutionary distance.

Click here for additional data file.

Figure S2 Metagenassist phenotype mapping of taxonomic classification based on 16S rRNA reads

The graphs show: energy sources, oxygen requirements, metabolism and biotic relationships.

Click here for additional data file.

Figure S3 SEM–EDS images of: (A) solid bed cut; (B) biofilm-rock interface; (C) minerals present in the rock-biofilm interface in the order of frequency

Click here for additional data file.

Figure S4 Venn diagram of common microbial families (total 214) identified by taxonomic analysis of amplicon sequencing of different groups

Click here for additional data file.

Table S1 PCR primers used in this study

Click here for additional data file.

Table S2 Protocols of PCR amplification of 16SrRNA

Click here for additional data file.

Table S3 Taxonomic distribution of sequencing reads obtained with archeal (Arch), actinobacterial (Act) and universal (Generic) primers

Click here for additional data file.

Table S4 Taxonomic distribution of OTU for 16S rRNA gene sequences obtained with archeal (Arch), actinobacterial (Act) and universal (Generic) primers

Length of colored data stripes is proportional to OTU’s number. (Clones) refers to data obtained from Tomczyk-Żak et al. (2013).

Click here for additional data file.

We gratefully acknowledge Professor MK Błaszczyk for his critical reading of the manuscript.

Additional Information and Declarations

Competing Interests

Author Contributions

DNA Deposition

The authors declare there are no competing interests.

Karolina Tomczyk-Żak performed the experiments, analyzed the data, prepared figures and/or tables, reviewed drafts of the paper.

Paweł Szczesny analyzed the data, contributed reagents/materials/analysis tools, wrote the paper, prepared figures and/or tables, reviewed drafts of the paper.

Robert Gromadka performed the experiments, reviewed drafts of the paper.

Urszula Zielenkiewicz conceived and designed the experiments, performed the experiments, analyzed the data, contributed reagents/materials/analysis tools, wrote the paper, prepared figures and/or tables.

The following information was supplied regarding the deposition of DNA sequences:

Sequence data from the GS FLX Titanium run has been deposited at the NCBI Short Read Archive (SRA) under project id of SRP093827.

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
