# Peer review of "Taxonomic and chemical assessment of exceptionally abundant rock mine biofilm"

_PeerJ, doi:10.7717/peerj.3635_

## Round 0.1 · original submission · Minor Revisions

Two of the reviewers suggested minor revisions, and one major revisions. All are enthusiastic about the paper, and so you should address all of the reviewers concerns. The paper has high value for the limits of life.

·

Basic reporting

The reporting is clear, fully referenced. The conclusions are well-founded.

Experimental design

The experimental design seems to be standard sequencing methodologies and is described in sufficient detail.

Validity of the findings

The findings are valid.

Additional comments

Comments and typos noted in the attached pdf file.

Reviewer 2 ·

Basic reporting

The paper by Tomczyk-Zak et al. explores the taxonomic diversity of a mine biofilm. The authors did an extensive work covering culturable (aerobic and anaerobic) and metagenomic amplicon analysis. They also performed an elemental analysis and suggested that silver and platinum inclusions could be the product of microbial metabolism. I think it is a valuable work, but it needs to be better described in some critical methodological aspects and tone down most of the metabolic predictions that the authors are claiming because they are over-reaching on several statements. Some results and methods are poorly described, please take note of the suggestions to improve your manuscript and give the detail to make their results reproducible. Please be extremely careful with your adjective use without describing the actual values (i.e. “The most highly, many sequences, etc.)

I would like you to think about a running title for your manuscript that describes your work fairly like:

“Microbial taxonomic and chemical diversity of a rock mine biofilm.”

Experimental design

This work is an original primary research. The authors are clear they are working with microbial diversity, but highly speculating about their results, which need to be toned down. The investigation is rigorous, but some technical standards are missing. Please take note about the comments about how to improve your methods.

Validity of the findings

Conclusions are not well supported by the presented evidence, particularly the model of “mineral’s cycling energy flow”, along with the “proposed biogeochemical transformation pathways”. This is speculation and is not identified as it, not as a part of the conclusion.

The authors should circumscribe themselves to just the data-supported conclusions (element analysis and microbial taxonomical diversity).

Additional comments

L49 What is Sub-aerial biofilm? I do not think there are flying biofilms or something similar.
L80 Is there a reason for alkaline pH here? Please define. Change case Alkaline to alkaline.
L95 Please define of your clonal analysis: what gene was analysed? Did you clone the genes? How many sequences? Change “the mine biofilm” for your studied mine name.
L95 remove “using a second generation of sequencing method” for pyrosequencing.
L124 Describe your DNA extraction briefly here, you are over-citing yourself along the text.
L175 What determined sequences? The sentence is not clear.
L187 “3% identity” Did you mean 97% identity? Or 3% dissimilarity? Please change the statement.
L187-188 Is there any amplicon size difference that could affect OTU clustering? If so, please declare the average length of each of your datasets.
L219 “actinobacteria” to Actinobacteria.
L246-L247 Please define inverse Simpson range to help readability (“the value”; what value?)
L253-254 Please add 0.03 dissimilarity. Add other diversity estimators that would help your readers to compare diversity to other microbial dominated ecosystems. Include: Shannon, Chao1 or ACE.
L259 change “Archaeal species” to Archaeal OTUs.
L269-L274. It is necessary to include how many sequences you are adding for each dataset (clones, universal, Archaea, Actinobacteria).
L277-L281 Looks like a grocery list. If you want to keep it, please include abundance information for each one of the phyla.
L281. “Many archeal”. How many archaeal?
L284. The most highly represented. How much?
L287. 8% of all reads? How many reads?
L289-L291. Include abundances for each proteobacteria listed: B-Proteobacteria (N, %N).
L295-L299. I do agree. However, in the same way it would be interesting to do Venn diagrams to illustrate the differences (qualitatively) due to primers selection.
L303. Eliminate or describe the actual values: “There is a substantial number of free-living…”
L304-L306. How did you determine that the microorganisms were connected to the nitrogen cycle? Add references on this topic.
L312-313. With taxonomic profile through 16S rRNA gene, you can not tell if there is a lack of microorganisms capable of iron redox reactions. Have you looked through the biochemistry? The gene expression? Metabolite intermediates? If don’t please remove your statement.
L316 contradicts what you just said in L312-313.
L321 change from “biofilm exhibits” to the biofilm microbiome taxonomic profile suggests a wide range…
L324-325 Are you comparing a Phylum to a Family? I think it is an unfair comparison.
L328-L330 Remove.
L333-334. Please define primers and amplicon average sequenced length in methods.
L342-345. How did you generate this phylogeny? Is it a precomputed RDP tree? Did you make it? Please elaborate on methods.
L347-349. It is not obvious, for me, I would remove the whole sentence it does not contribute, and it is not well written or with supporting evidence.
L351. The conditions are not clear. Please define here.
L352. You should abide the pan-genomic variability and that even two strains of the same species could have extremely divergent traits like being pathogenic or commensal, think about E. coli. They are the same using 16S rRNA gene, but phenotypes could be contrasting. So, try to delimitate this to predicted phenotypic traits and make a statement of the possible drawbacks using metagenassist.
L395 Why is the environment oligotrophic? Which is the limiting nutrient? Are there measurements of nutrients here? Do you have values?
L399. Microflora is for tiny plants. Microbiota is for microorganisms.
L399. Do you have any support to define indigenous microbiota?
L405. References for other tailings, caves, etc. actual values for alpha diversity here, you are not comparing a single value here. For example, the cave has a 8.0 Shannon, while tailings have 3.0, acid mine drainage 2.0 (just imaginary values here). I want real diversity values for this statement.
L409. “substantially enriching” Can you put a number on this?
L411 “… last few years.” Do you have any references supporting your statement?
L411 What is high biodiversity?
L413 What is your restrictive approach?
L418 remove species change for strains.
L420 Significant? Statistical? What alpha did you consider? What is the difference?
L423. Maybe a rank abundance curve included as a figure could help with this last statement.
L424. You can not predict genome size straight forward from 16S please include that this is just a coarse speculation.
L428. “small number” what is it?
L431 What three kind of interaction?
L434 energy sources, do you have measurable data on this? If not, please define as probable nitrogen compounds.
L 441 Can you mention some examples of phenotypic databases?
L442 What capabilities? I ready your 2013 and 2012 paper and you did not have support for this statement, you can not just jump from 16S profiling to metabolic predictions, and then citing them. You did some work for arsenic just amplifying the genes but not showing they express themselves or translate them.
L446-L450. Please clarify what evidences you have for the shown metabolic functions. It is okay to speculate; you have a sound basis for it. But just make it clear and separate it from the evidence.
L483. What do refer to Gauss law? The electric one?
L486. Do you have any reference for the plankton paradox? Please include it
L504. I would change it from proposed to predicted, and please declare how you made your prediction (metagenassist), consider discussing with similar strategies like picrust, and summarize the limitations (remember the E. coli example mentioned before).
L511-514 are speculation, and you did not measure the microbial metabolic activity.

·

Basic reporting

Article review
Taxonomic and chemical assessment of exceptionally abundant rock mine biofilm.

The article entitled "Taxonomic and chemical assessment of exceptionally abundant rock mine biofilm" describes the biofilm present in rocks of the Zloty Stok gold and arsenic mine. The description is made through taxonomic, microorganisms isolation and rock chemical analyzes.
The authors report a biofilm with a very high biodiversity studied through sequencing of 16s rRNA gene amplicons obtained from generic primers for both archaea and actinobacteria.
Orthography
Line 72: 2003),which. Is neccesary a space between “,” and “which”.
Line 224: SEM methodology showed in → Shown en vez de showed
Line 434: 2013).The major. Is neccesary a space between “.” and “The”
Article
The Figure 2 "Diagrammatic cross-section of the rock biofilm simple test" does not provide any extra information that helps to have a clearer idea of the cuts that were made for the later analyzes of the Biofilm and the rock in which it is attached. If there is no more representative scheme, it is suggested to take out the image of the article and only leave the explanation: Further analyzes were performed on three distinct layers of the sample - the biofilm (layer 0), an interface between biofilm and underlying rock (layer 1), and finally the solid bed under the biofilm (layer 2: see Fig. 2 schematic).
--
I
--

The authors in their discussion mention that since arsenic has a uniform distribution in the biofilm, it is possible that the microorganisms present do not use arsenic compounds as sources of energy. This they support it based on its results that within the 52 strains that isolated, none was able to obtain energy from arsenic compounds.
In spite of presence of arsenic oxidation and reduction genes, besides of the antecedent of another group where a strain that possesses the mentioned genes was isolated (Drewniak et al., 2010, 2012; 2014), the authors affirm that they do not believe that this metabolism is happening in the rock. Independent of what the authors believe, with the results of this article cannot determine if the species that inhabit the biofilm rock use or not arsenic compounds as sources of energy.

In several sections of the article more references are needed, especially in the discussion (and conclusion when they mention it the plankton paradox).

In my view, the article should be accepted with minor corrections as far as spelling and writing. In addition, discussion and conclusion should be developed in greater detail. It is advisable to carry out comparative analyzes between the relative abundances of archaea, antibacterial and other taxonomic groups to give more strength to the results.

Experimental design

t is a fact that the sequencing with different primers introduces a bias when comparing the structure of the studied communities. The authors mention that because of this, they did not attempt to perform a statistical analysis to determine differences in the abundance of the taxonomic groups (295 to 298 lines). However, there are some methodologies to compensate the differences in depth of sequencing, as well as different uses of primers, among others. Articles describing this type of analysis are detailed below.

Leek, J. T., Scharpf, R. B., Bravo, H. C., Simcha, D., Langmead, B., Johnson, W. E., ... & Irizarry, R. A. (2010). Tackling the widespread and critical impact of batch effects in high-throughput data. Nature Reviews Genetics, 11(10), 733-739.

Leek, J. T. (2014). svaseq: removing batch effects and other unwanted noise from sequencing data. Nucleic acids research, gku864.

Love, M. I., Huber, W., & Anders, S. (2014). Moderated estimation of fold change and dispersion for RNA-seq data with DESeq2. Genome biology, 15(12), 550.

McMurdie, P. J., & Holmes, S. (2013). phyloseq: an R package for reproducible interactive analysis and graphics of microbiome census data. PloS one, 8(4), e61217.

Validity of the findings

In summary, this work provides a new information about the microorganism communities inhabiting the biofilm present in rocks of the Zloty Stok mine in Poland. The authors show that this community has a very high biodiversity, a result that would describe (according to the authors) a behavior such as plankton paradox. In addition, the mineralogy of the rock is analyzed and with these results a correlation with the biogeochemical processes produced by bacteria is established to finally present possible ways of mineralogical transformation.

Additional comments

Article review
Taxonomic and chemical assessment of exceptionally abundant rock mine biofilm.

The article entitled "Taxonomic and chemical assessment of exceptionally abundant rock mine biofilm" describes the biofilm present in rocks of the Zloty Stok gold and arsenic mine. The description is made through taxonomic, microorganisms isolation and rock chemical analyzes.
The authors report a biofilm with a very high biodiversity studied through sequencing of 16s rRNA gene amplicons obtained from generic primers for both archaea and actinobacteria.
Orthography
Line 72: 2003),which. Is neccesary a space between “,” and “which”.
Line 224: SEM methodology showed in → Shown en vez de showed
Line 434: 2013).The major. Is neccesary a space between “.” and “The”
Article
The Figure 2 "Diagrammatic cross-section of the rock biofilm simple test" does not provide any extra information that helps to have a clearer idea of the cuts that were made for the later analyzes of the Biofilm and the rock in which it is attached. If there is no more representative scheme, it is suggested to take out the image of the article and only leave the explanation: Further analyzes were performed on three distinct layers of the sample - the biofilm (layer 0), an interface between biofilm and underlying rock (layer 1), and finally the solid bed under the biofilm (layer 2: see Fig. 2 schematic).
--
It is a fact that the sequencing with different primers introduces a bias when comparing the structure of the studied communities. The authors mention that because of this, they did not attempt to perform a statistical analysis to determine differences in the abundance of the taxonomic groups (295 to 298 lines). However, there are some methodologies to compensate the differences in depth of sequencing, as well as different uses of primers, among others. Articles describing this type of analysis are detailed below.

Leek, J. T., Scharpf, R. B., Bravo, H. C., Simcha, D., Langmead, B., Johnson, W. E., ... & Irizarry, R. A. (2010). Tackling the widespread and critical impact of batch effects in high-throughput data. Nature Reviews Genetics, 11(10), 733-739.

Leek, J. T. (2014). svaseq: removing batch effects and other unwanted noise from sequencing data. Nucleic acids research, gku864.

Love, M. I., Huber, W., & Anders, S. (2014). Moderated estimation of fold change and dispersion for RNA-seq data with DESeq2. Genome biology, 15(12), 550.

McMurdie, P. J., & Holmes, S. (2013). phyloseq: an R package for reproducible interactive analysis and graphics of microbiome census data. PloS one, 8(4), e61217.

--

The authors in their discussion mention that since arsenic has a uniform distribution in the biofilm, it is possible that the microorganisms present do not use arsenic compounds as sources of energy. This they support it based on its results that within the 52 strains that isolated, none was able to obtain energy from arsenic compounds.
In spite of presence of arsenic oxidation and reduction genes, besides of the antecedent of another group where a strain that possesses the mentioned genes was isolated (Drewniak et al., 2010, 2012; 2014), the authors affirm that they do not believe that this metabolism is happening in the rock. Independent of what the authors believe, with the results of this article cannot determine if the species that inhabit the biofilm rock use or not arsenic compounds as sources of energy.

In several sections of the article more references are needed, especially in the discussion (and conclusion when they mention it the plankton paradox).

In summary, this work provides a new information about the microorganism communities inhabiting the biofilm present in rocks of the Zloty Stok mine in Poland. The authors show that this community has a very high biodiversity, a result that would describe (according to the authors) a behavior such as plankton paradox. In addition, the mineralogy of the rock is analyzed and with these results a correlation with the biogeochemical processes produced by bacteria is established to finally present possible ways of mineralogical transformation.

In my view, the article should be accepted with minor corrections as far as spelling and writing. In addition, discussion and conclusion should be developed in greater detail. It is advisable to carry out comparative analyzes between the relative abundances of archaea, antibacterial and other taxonomic groups to give more strength to the results.

---

## Round 0.2 · accepted · Accept

The authors performed all the editorial changes requested by both reviewers and now it is less speculative and it reads well.